# Animal Video Lovers Always Have Company: The Role of Cyber-Mediated Animal Attachment in Loneliness

**DOI:** 10.3390/ani15172593

**Published:** 2025-09-04

**Authors:** Junzi Zhang, Su Tao, Wenchong Du

**Affiliations:** 1School of Marxism, China University of Geosciences (Beijing), Beijing 100083, China; zhangjunzi@email.cugb.edu.cn; 2NTU Psychology, School of Social Science, Nottingham Trent University, Nottingham NG1 4FQ, UK

**Keywords:** animal video engagement, cyber-mediated animal attachment, loneliness, animal attachment, parasocial attachment

## Abstract

Many people enjoy watching cute or heartwarming animal videos online, but little is known about how this activity may affect our emotions. This study explored whether watching animal videos on the internet can help reduce feelings of loneliness. The researchers introduced a new concept called “cyber-mediated animal attachment,” which refers to the emotional connection people can form with animals they see in digital media. Using a survey and an experiment, this study found that people who spent more time watching animal videos tended to feel less lonely. This was not just because the videos were fun or uplifting. Instead, it was the emotional bond with the animals that played a key role in reducing loneliness. Compared with funny videos featuring people, animal videos were more likely to create such emotional connections. These findings suggest that animals, even when viewed through a screen, can provide a sense of connection. This highlights the unique role of animals in supporting emotional well-being and suggests potential ways in which digital animal content might help people.

## 1. Introduction

In the age of short-video platforms, animal-related content has become one of the internet’s most beloved genres. From playful puppies to majestic wildlife, a wide variety of animals now dominate digital spaces, with their videos often going viral and transforming into memes or widely shared clips [1,2]. This phenomenon has given rise to a new form of digital behavior—*online animal video engagement*—which includes watching, liking, sharing, and commenting on animal content across social media and streaming platforms. Among the most active participants in this trend are young people [3,4], who not only consume these videos frequently but, in some cases, develop deep emotional bonds, becoming loyal followers or even devoted fans of specific animal “influencers” [5]. However, beyond brief entertainment, the psychological implications of these interactions remain underexplored.

While extensive research highlights that animal companions can serve as substitutes for lacking social support [6,7] and help reduce feelings of loneliness [8,9], it remains unclear whether similar outcomes can be achieved through virtual, non-physical interactions with animals. Given the global rise in loneliness and social disconnection, especially among youth and digitally connected populations [10], it is crucial to investigate whether digital media can foster meaningful emotional connections that buffer against loneliness.

Previous studies have shown that indirect interactions—such as through imagination, being in pet-related environments, or recalling pet-related experiences—have been associated with higher psychological well-being [11]. Theoretical frameworks such as the Media Equation Theory suggest that individuals tend to respond to media as if they were real social entities [12], with media interactions capable of eliciting physiological and emotional responses similar to those evoked by real-life stimuli [13,14,15]. These perspectives raise the following question: can viewers develop attachment-like bonds with animals in online videos, and might these bonds have measurable psychological benefits?

To address this, we introduce the concept of cyber-mediated animal attachment—an emotional connection formed with animals featured in digital media. Unlike traditional pet attachment, which relies on physical interaction, cyber-mediated animal attachment arises entirely through repeated exposure to and engagement with animal video content. We propose that this type of virtual bond may play a role in reducing loneliness, functioning similarly to real-life pet attachment in offering emotional support and perceived social connectedness.

This study aims to fill two key gaps in the literature: first, by empirically testing whether online animal video engagement is associated with loneliness, and second, by examining whether cyber-mediated animal attachment mediates this relationship. Across two complementary studies, we also assess whether the effect is distinct from general positive emotional responses and whether it holds beyond stable personality traits. In doing so, we aim to clarify the psychological value of virtual human–animal connections in an increasingly digital world.

### 1.1. The Positive Effects of Watching Animal Videos

Research on animal video watching has primarily explored its positive effects, which can be categorized into three main aspects. First, animal videos often win viewers’ affection with their cute appearances and humorous behaviors, which can quickly improve the viewers’ mood. For instance, watching cat videos online has been shown to be positively correlated with positive emotions and negatively correlated with negative emotions [16]; dog videos were found to enhance positive emotional responses more effectively than nature or control videos [17]; and animal videos shown during university break times significantly improved students’ moods compared to no videos at all [18]. Second, watching animal videos can alleviate anxiety and stress. Studies have demonstrated that dog videos reduce subjective anxiety more effectively than nature or control videos [17], and images or videos of animals can serve as substitutes for therapeutic animals to relieve stress, leading to reduced physiological stress responses [19,20,21,22]. Third, watching animal videos has a positive impact on individuals’ well-being. Pet videos and livestreams can induce a state of focused and pleasurable flow experience, which, in turn, enhances subjective well-being [23]. Both calm and active interactions with animals have been shown to increase happiness [20]. While prior work has demonstrated the mood-enhancing and stress-reducing effects of animal videos, the potential of such content to alleviate deeper social–emotional challenges—such as loneliness—has received limited empirical attention. Accordingly, we propose the following hypothesis:

**Hypothesis** **1.**
*Online animal video engagement negatively associates with loneliness.*


### 1.2. Definition and Effect of Cyber-Mediated Animal Attachment

Although prior research attributes the benefits of animal videos to their capacity to elicit positive emotions and reduce stress, this explanation does not sufficiently account for their unique psychological impact. Other forms of content, such as humorous human videos, can generate similar emotional effects. What may distinguish animal videos is their potential to evoke emotional bonding, a deeper form of psychological engagement. Mood Management Theory (MMT) posits that individuals use media with the motivation to manage their emotions, either to eliminate unpleasant emotional states or to maintain positive ones. But, not all emotional management behaviors are aimed at pleasure; some viewers prefer emotionally meaningful media content because it fosters psychological connections with others [24]. Animals, as typical emotional companions, demonstrate unique potential in such connections.

Poresky and Hendrix [25] distinguished between the psychological implications of pet presence and pet bonding, finding that emotional support and psychological satisfaction are obtained only when an attachment with the animal is established. Owning a companion animal was not significantly associated with any psychological well-being indicators; rather, attachment to the animal emerged as a robust predictor of mental health outcomes [26]. We argue that analogous psychological processes may occur in the digital realm, whereby viewers develop affective ties to animals they encounter repeatedly through online content, even in the absence of physical interaction. Parasocial Relationships theory [27] supports the notion that individuals can form one-sided but emotionally meaningful bonds with media figures. Extending this framework, we posit that similar parasocial processes can occur with animal figures in video content, giving rise to a form of digital attachment. Through repeated exposure to online animal content, viewers may develop comparable emotional bonds. Based on this premise, we introduce the concept of *cyber-mediated animal attachment*, referring to a novel form of animal attachment shaped by parasocial intimacy via social media platforms. Unlike anthropomorphism toward fictional characters or influencers, cyber-mediated animal attachment may be uniquely grounded in perceived innocence, warmth, and non-verbal emotional expression characteristic of animals—features that may foster empathic responses and feelings of companionship [28,29]. Unlike traditional pet attachment, which is grounded in actual pet ownership, cyber-mediated animal attachment may occur both among individuals who own animals and those with no prior pet-owning experience.

Secure attachment has been shown to alleviate feelings of loneliness [30,31], and attachment to animals has been found to rival or even exceed attachment to humans in its psychological impact [32]. According to the Media Equation Theory [12], we anticipate that cyber-mediated animal attachment may yield loneliness-reducing effects similar to those of real-life pet attachment. Several lines of reasoning support this hypothesis.

First, within the framework of attachment theory, owners and pets can regard each other as sources of “safe havens” and “secure bases” [33,34,35]. Zasloff and Kidd [6] posited that pet attachment can serve as a substitute for social support under certain conditions. When interpersonal functioning is impaired or loneliness is experienced, individuals may turn to animals to compensate for relational deficits. Indeed, animal ownership—compared to non-ownership—was associated with smaller increases in loneliness following the COVID-19 lockdown [36]. Although online animals cannot provide physical intimacy, the notion of “petness” as a social construct is not confined to live animals in physical proximity; it can also be projected onto inanimate objects and distant sentient beings [37]. Both real and virtual animals can serve as emotional aides and enhancers of well-being [38].

Second, animal attachment has been identified as a key mechanism for stress relief and emotional regulation. Stronger attachment to pets has been associated with better mental health [39], and the human–animal bond has been shown to buffer the impact of stress and adversity, especially among individuals with limited social support [40]. Similarly, pet affinity has been found to moderate the relationship between ambivalence over emotional expression (AEE) and perceived social support, serving as a protective factor in emotional well-being [41]. Krause-Parello [8] further conceptualized loneliness itself as a source of stress and argued that animal attachment serves as a moderating mechanism that buffers the relationship between loneliness and physical health. Previous findings on the stress-reducing and mood-enhancing effects of animal videos and images lend further support to this mechanism [16,17,21].

Third, humans have a natural tendency toward anthropomorphism—readily attributing intentionality and mental states to animals [28]. Companion animal owners frequently ascribe emotional states to their pets [42], and individuals are capable of recognizing subtle emotional expressions in animals, such as cats’ facial cues [43]. There is a documented association between anthropomorphism and pet attachment [44]; owners who anthropomorphize their pets score higher on the Pet Bonding Scale [42]. Notably, such anthropomorphic interpretations can be triggered simply by an animal’s image or video, suggesting that meaningful psychological engagement can occur even in virtual contexts.

Based on this, even in the absence of physical interaction, viewers may still form cyber-mediated animal attachment, which can serve as a significant factor in reducing loneliness. Therefore, this study proposes the following hypothesis:

**Hypothesis** **2.**
*Cyber-mediated animal attachment mediates the relationship between online animal video engagement and loneliness.*


### 1.3. Overview of This Study

This study employs two sub-studies to test the proposed mediation model and to rule out alternative explanations.

Study 1 employed a questionnaire survey to investigate the relationships among online animal video engagement, cyber-mediated animal attachment, and loneliness. While a negative association between video engagement and loneliness was hypothesized, it is also plausible that frequent—or even compulsive—engagement with short animal videos could be linked to increased feelings of loneliness. Prior research has consistently demonstrated a close association between excessive internet use and heightened loneliness. For example, heavy internet reliance has been found to reduce real-life social interaction [45] and contribute to emotional loneliness [46,47]. Problematic internet use (PIU) and loneliness appear to form a vicious cycle, with PIU potentially serving as the initial trigger [48]. Online animal video engagement often takes place on short-video platforms, which have been associated with addictive use patterns and adverse psychological outcomes. Studies have linked short-video addiction to increased social exclusion [49] and depressive symptoms [50]. Addictive users of short-video apps tend to report worse mental health conditions than non-users and moderate users, including greater loneliness [51]. Additionally, some users watch cat videos to procrastinate, sometimes resulting in guilt [16]. These findings suggest that excessive online animal video engagement may be accompanied by risks commonly associated with problematic internet use and short-video addiction. Therefore, to assess the possibility of such negative effects, Study 1 also tested the potential nonlinear relationship between online animal video engagement and loneliness.

Stable personality factors are among the most important predictors of individual differences in loneliness [52], and the relationship between personality and loneliness has been shown to be substantially genetic in nature [53]. To establish the added predictive value of online engagement and cyber-mediated animal attachment, we statistically control for established individual difference variables [54]. Therefore, in Study 1, the Big Five traits were included as control variables to test the incremental validity of online animal video engagement and cyber-mediated animal attachment in predicting loneliness beyond personality.

Study 2 employed an experimental design to rule out the alternative explanation that the reduction in loneliness associated with watching animal videos is due to positive emotions. To control for the immediate emotional impact of video content, this study used animal videos and humorous human videos that elicited comparable levels of positive affect. This design enabled us to isolate the effect of cyber-mediated animal attachment from that of general positive affect, providing a more rigorous test of its unique mediating role between online animal video viewing and loneliness.

## 2. Study 1: The Association of Online Animal Video Engagement, Cyber-Mediated Animal Attachment, and Loneliness

### 2.1. Pre-Study: Questionnaire Development and Revision

In this study, we developed original scales for online animal video engagement and cyber-mediated animal attachment. The development process followed standard psychometric procedures:

First, through a literature review and individual interviews, we identified the specific behaviors related to online animal video engagement and explored the dimensions and nature of cyber-mediated animal attachment. Next, we drafted the items for the scales. The online animal video engagement scale was primarily based on behavioral patterns identified through the interviews. The cyber-mediated animal attachment scale was based on the Lexington Attachment to Pets Scale (LAPS) developed by Johnson et al. [55], with references to “pets” modified to refer to “online animals,” and additional items incorporated based on insights from participant interviews. Third, expert reviews and pilot testing were conducted, leading to revisions in the wording and content of the items. Fourth, Sample 1 (154 participants) was collected for preliminary item analysis and exploratory factor analysis. Finally, Sample 2 (416 participants) was collected for structural validation of the questionnaire.

SPSS 26.0 was used to perform item analysis and exploratory factor analysis on the Sample 1 data. In the preliminary online animal video engagement questionnaire (seven items), one item was removed after item analysis. Exploratory factor analysis revealed a single factor, explaining 63.68% of the variance, resulting in a final scale with six items. The initial cyber-mediated animal attachment questionnaire consisted of 26 items. After item analysis, five items were removed, and exploratory factor analysis identified three factors in the cyber-mediated animal attachment scale: emotional attachment (emotional responses and happiness derived from online animals), importance attachment (the importance of online pets in one’s life), and intimacy attachment (the sense of closeness and emotional bond with online pets), which together explained 64.35% of the total variance, resulting in a final scale with 12 items.

Internal consistency testing and confirmatory factor analysis were conducted using SPSS 26.0 and Mplus 8.0 on the Sample 2 data. Cronbach’s *α* was 0.819 for the online animal video engagement scale and 0.835 for the cyber-mediated animal attachment scale. The model fit indices from confirmatory factor analysis were as follows: for the online animal video engagement scale, *χ*^2^/*df* = 4.43, *CFI* = 0.959, *TLI* = 0.932, and *RMSEA* = 0.091; for the cyber-mediated animal attachment scale, *χ*^2^/*df* = 2.77, *CFI* = 0.935, *TLI* = 0.915, and *RMSEA* = 0.065. Both scales demonstrated acceptable model fit. These two scales were subsequently used in the main studies. The scale development process and detailed data are provided in Appendix A.

### 2.2. Materials and Methods

#### 2.2.1. Participants

Participants were recruited via university mailing lists and social media platforms using convenience sampling. Eligible participants were required to be over 18 years old and active users of short-video platforms. Data were collected through both online and offline surveys, resulting in 299 valid responses. Of these, 141 participants were male (47.2%) and 158 were female (52.8%). A total of 61 participants (20.4%) had currently or previously owned real pets. The mean age was 23.11 ± 1.44 years.

A post hoc power analysis [56] using G*Power 3.1 indicated that with a sample size of 299, an alpha level of 0.05, and a desired power of 0.80, this study was capable of detecting a minimum effect size of *r* = 0.08 (two-tailed).

#### 2.2.2. Measures

*Online Animal Video Engagement.* Online animal video engagement was assessed using a self-developed 6-item scale. Example items include “How frequently do you watch animal videos?” and “How frequently do you share your favorite online animal videos with others?” Responses were rated on a 5-point Likert scale (1 = never, 2 = rarely, 3 = occasionally, 4 = sometimes, and 5 = often). In the present study, the scale demonstrated good internal consistency, with a Cronbach’s *α* of 0.853.

*Cyber-mediated Animal Attachment*. Cyber-mediated animal attachment was measured using a self-developed 12-item scale. Example items include “Animal videos are indispensable in my life” and “I feel a close bond with my favorite online animals.” Participants responded on a 5-point Likert scale (1 = completely inconsistent; 5 = completely consistent). In the present study, the scale demonstrated good internal consistency, with a Cronbach’s *α* of 0.847.

*Loneliness.* Loneliness was assessed using the UCLA Loneliness Scale (ULS), developed by Russell [57]. The scale consists of 20 items, including 9 reverse-scored items, rated on a 4-point Likert scale (1 = never, 2 = rarely, 3 = sometimes, and 4 = always). In this study, Cronbach’s *α* was 0.904.

*Personality Traits.* Personality traits were assessed using the Ten-Item Personality Inventory (TIPI) [58], a brief measure of the Big Five personality traits. The scale comprises ten items, with two items representing each of the five dimensions. Responses were recorded on a 5-point Likert scale (1 = completely inconsistent; 5 = completely consistent), with five items reverse-scored. The psychometric properties of the scale have been validated across different studies [59,60].

#### 2.2.3. Procedure

Participants sequentially completed the scales measuring online animal video engagement, cyber-mediated animal attachment, loneliness (ULS), and personality traits (TIPI). They then provided their demographic information. Data were analyzed using SPSS 26.0, including descriptive statistics, correlation analysis, regression analysis, and mediation model testing. This study received ethical approval from the Psychological Ethics Committee of the first author’s institution, and all participants provided informed consent prior to participation.

### 2.3. Results

#### 2.3.1. Common Method Bias Test

Since all data in this study were collected through self-reported questionnaires, to assess common method bias, we conducted Harman’s single-factor test. The results of the exploratory factor analysis indicated that 11 factors with eigenvalues greater than 1 were extracted, with the first factor accounting for 21.06% of the total variance (which is below the 40% threshold). Therefore, common method bias does not pose a serious concern in this study.

#### 2.3.2. Descriptive Statistics and Correlation Analysis of Variables

Pearson correlation analyses were conducted for online animal video engagement, cyber-mediated animal attachment, loneliness and personality traits (see Table 1). The results indicated a significant negative correlation between online animal video engagement and loneliness. Cyber-mediated animal attachment was positively correlated with online animal video engagement and negatively correlated with loneliness, suggesting its potential as a mediator. Additionally, loneliness showed significant positive correlations with all five dimensions of the Big Five personality traits. Therefore, personality traits were included as control variables in subsequent mediation analyses.

#### 2.3.3. Nonlinear Relationship Between Online Animal Video Engagement and Loneliness

To examine whether a nonlinear relationship exists between online animal video engagement and loneliness, we standardized the online animal video engagement variable and created a squared term (Quadratic). A regression analysis was conducted with both online animal video engagement and Quadratic as predictors, and loneliness as the dependent variable. The model yielded an *R*^2^ = 0.065, *F* = 10.21, *p* < 0.001. The results indicated that online animal video engagement negatively predicted loneliness (*β* = −0.23, *t* = −4.14, *p* < 0.001), whereas the Quadratic term was not statistically significant (*β* = −0.10, *t* = −1.74, *p* = 0.083). These findings suggest that the relationship between online animal video engagement and loneliness is linear rather than nonlinear (see Figure 1).

#### 2.3.4. Relationships Among Online Animal Video Engagement, Cyber-Mediated Animal Attachment, and Loneliness

After centering all continuous variables, Model 4 of the PROCESS macro was used to examine the mediating effect, with online animal video engagement as the independent variable, cyber-mediated animal attachment as the mediating variable, and loneliness as the dependent variable. Age, gender, pet ownership, and personality traits were included as control variables. The results are presented in Table 2. The results of a nonparametric Bootstrap test indicated a significant mediating effect of cyber-mediated animal attachment (*total effect* = −0.05, *SE* = 0.03, 95% *CI* [−0.100, −0.001]; *indirect effect* = −0.03, *SE* = 0.02, 95% *CI* [−0.068, −0.003]). The indirect effect accounted for 60% of the total effect. Upon incorporating cyber-mediated animal attachment into the regression model, the direct effect of online animal video engagement on loneliness became non-significant (*direct effect* = −0.02, *SE* = 0.03, 95% *CI* [−0.075, −0.044]), suggesting that cyber-mediated animal attachment fully mediated this relationship, thus supporting Hypothesis 2. The standardized path coefficients are illustrated in Figure 2.

Additionally, personality traits were included as covariates in the model. Extraversion and agreeableness negatively predicted loneliness, whereas neuroticism positively predicted loneliness. Importantly, even after accounting for personality traits, cyber-mediated animal attachment significantly predicted loneliness, suggesting that it explained variance beyond that accounted for by personality traits, thereby demonstrating incremental validity.

## 3. Study 2: The Impact of Watching Animal Videos on Attachment and State Loneliness

In addition to animal videos, short videos commonly found online include comedy, educational and popular science content, lifestyle vlogs, cooking tutorials, and current affairs. Many of these videos can trigger positive emotional responses. To isolate the unique psychological effects of animal videos, this study uses humorous human videos with clear main characters that evoke similar levels of positive emotions. Participants’ attachment to these characters was measured to test whether cyber-mediated animal attachment influences viewers’ state loneliness beyond the effect of positive emotions. This study proposes the following hypothesis: Compared to those who watch humorous human videos, individuals who watch animal videos will report stronger attachment to the video character and lower levels of state loneliness.

### 3.1. Materials and Methods

#### 3.1.1. Participants

Sample size was estimated using G*Power 3.1, with an effect size of 0.5, an *α* level of 0.05, and a power (1 − *β*) of 0.80. The calculation indicated that 64 participants were needed per group, for a total of 128 participants. Data were collected via the Wenjuanxing platform (www.wjx.com), resulting in 131 valid responses. Among the participants, 65 were male and 66 were female, with a mean age of 27.43 ± 4.84 years. A total of 26 participants (19.8%) had currently or previously owned real pets.

#### 3.1.2. Materials and Measures

*Attachment to Video Character*. The cyber-mediated animal attachment scale was adapted to suit different types of video content. Specifically, references to “animal videos” were modified to “this video”, such as changing “I feel happy when watching animal videos” to “I feel happy when watching this video.” In the humorous human video condition, references to “animals” were changed to “people”, for example, “I feel close to the animals in the videos” was modified to “I feel close to the people in this video.” In this study, Cronbach’s *α* was 0.833.

*Baseline Loneliness*. RULS-6 [61] was employed to assess participants’ baseline level of loneliness. A sample item is “How often do you feel that you lack companionship?” Responses were rated on a 4-point Likert scale. In this study, Cronbach’s *α* was 0.842.

*State Loneliness*. Participants rated the extent to which they currently felt lonely using the ULS-8 [62]. Both the ULS-8 and RULS-6 are derived from the ULS, with three items overlapping between the two versions. To measure state loneliness, the instructions were modified to prompt participants to reflect on their feelings “at this very moment.” A sample item is “Right now, I feel left out.” Responses were recorded on a 4-point Likert scale (1 = Not at all; 4 = Very much). In this study, Cronbach’s α was 0.780.

*Video Materials*. Ten high-viewership videos were selected from the Chinese short-video platform Bilibili (www.bilibili.com), including five animal videos and five humorous human videos. Each video was approximately 1 min long and easily understandable without explanation.

To select videos eliciting comparable levels of positive emotions, 40 participants (20 male and 20 female) were recruited to evaluate the emotional impact of the videos. Participants randomly viewed videos of different types and assessed their immediate emotional reactions after each video using the Differential Emotions Scale (DES) [63]. After a five-minute break, they viewed the next video and completed the scale again. Each participant evaluated five videos presented in a random order.

The items for relaxation, enjoyment, and happiness in the DES were combined into a positive emotion dimension. Based on the ratings, one animal video and one humorous human video that elicited similar levels of positive emotions were selected for the main study. The positive emotion rating for the selected animal video was *M* = 13.80, *SD* = 4.68; for the humorous human video, *M* = 13.75, *SD* = 6.56.

#### 3.1.3. Research Design and Procedure

Participants were randomly assigned to one of two conditions (animal video or humorous human video) in a between-subjects design. They first completed the RULS-6 to assess their baseline level of loneliness. Next, they watched the assigned video and then sequentially completed the attachment scale, the ULS-8, and rated their perceived positive emotions (happiness and joy) on a 5-point Likert scale. Finally, participants provided demographic information.

### 3.2. Results

#### 3.2.1. Effects of Video Type on Loneliness, Attachment to Video Characters, and Positive Emotions

Independent samples *t*-tests were conducted to compare differences between the animal video group and the humorous human video group in terms of baseline and state loneliness, attachment, and positive emotions. For baseline loneliness, there was no significant difference between participants in the animal video group (*M* = 2.04, *SD* = 0.56) and those in the humorous human video group (*M* = 2.09, *SD* = 0.64), *t* (129) = 0.57, *p* = 0.569, indicating that the group assignment was unbiased. For state loneliness, participants in the animal video group (*M* = 1.90, *SD* = 0.50) reported significantly lower levels than those in the humorous human video group (*M* = 2.13, *SD* = 0.36), *t* (126) = 2.99, *p* = 0.003, *d* = 0.53. For attachment to video characters, participants in the animal video group (*M* = 3.55, *SD* = 0.60) reported significantly higher attachment than those in the humorous human video group (*M* = 2.94, *SD* = 0.71), *t* (129) = −5.29, *p* < 0.001, *d* = 0.93. The ratings of “happy” and “joyful” were highly correlated (*r* = 0.80, *p* < 0.001), so their mean score was used as the index of positive emotions. No significant difference was found between the animal video group (*M* = 4.86, *SD* = 1.17) and the humorous human video group (*M* = 4.98, *SD* = 1.19) in positive emotions, *t* (129) = 0.55, *p* = 0.583.

#### 3.2.2. Mediation Analysis of Attachment to Video Characters

To examine whether attachment to video characters mediated the relationship between video type and loneliness, while controlling for the potential mediating effect of positive emotion, a mediation analysis was conducted using PROCESS macro Model 4 in SPSS 26.0. All continuous variables were mean-centered prior to analysis. Loneliness served as the dependent variable, while video type (animal vs. human) as the independent variable. Attachment and positive emotion were included as mediators, and age, gender, pet ownership, and baseline loneliness were considered as control variables. The results are presented in Table 3. The results of a bootstrap non-parametric test indicated that the total effect of video type on loneliness was significant (*total effect* = −0.210, *SE* = 0.06, 95% *CI* [−0.333, −0.088]). The direct effect was not significant after the inclusion of the mediators (*direct effect* = −0.114, *SE* = 0.07, 95% *CI* [−0.245, 0.018]). Furthermore, the indirect effect through attachment was significant (*effect* = −0.098, *SE* = 0.03, 95% *CI* [−0.170, −0.039]), while the indirect effect through positive emotion was not significant (*effect* = 0.002, *SE* = 0.01, 95% *CI* [−0.014, 0.018]). The indirect effect via attachment accounted for 46.67% of the total effect. The standardized mediation path diagram is provided in Figure 3.

## 4. Discussion

### 4.1. Main Findings of This Study

This study explored whether emotional attachment to animals featured in online videos—termed cyber-mediated animal attachment—can serve as a buffer against loneliness. Across two studies, we found that online animal video engagement was negatively associated with loneliness and that this relationship was mediated by cyber-mediated animal attachment. These results advance the existing literature by identifying a specific psychological mechanism, emotional bonding with virtual animals, that underpins the mental health benefits of animal content in digital media.

The results showed that online animal video engagement significantly negatively predicted loneliness. Compared to individuals who rarely engage with such content, those who frequently watch online animal videos and interact through liking, commenting, and sharing reported significantly lower levels of loneliness. Furthermore, no significant nonlinear relationship was found between online animal video engagement and loneliness, indicating that frequent viewing does not pose a risk of short-video addiction. These findings support Hypothesis 1 and are consistent with previous studies suggesting the positive psychological outcomes of watching animal videos [16,17,21]. In the mediation model of Study 1, the direct effect of online animal video engagement on loneliness was not significant, whereas the indirect effect via cyber-mediated animal attachment was significant, indicating full mediation and supporting Hypothesis 2. This suggests that it is not video engagement per se that reduces loneliness, but the emotional attachment individuals develop toward the featured animals. Such attachment, akin to real-life pet bonding, appears to be the mechanism through which psychological benefits emerge. This finding aligns with that of Wells, Clements, Elliott, Meehan, Montgomery, and Williams [26], who argued that the emotional benefits of animal interactions depend on attachment rather than mere exposure or ownership. Therefore, when online interactions with animals evoke a sense of intimacy, empathy, or irreplaceability, cyber-mediated animal attachment may provide emotional value and social connectedness, thereby alleviating loneliness. Importantly, this mediating effect remained significant even after controlling for personality traits and other known predictors of loneliness. This indicates that cyber-mediated animal attachment has unique and incremental explanatory power, highlighting its essential role in fostering psychological connections in the digital media age.

To investigate whether cyber-mediated animal attachment exerts effects independent of positive affect, Study 2 compared animal videos with humorous human videos. These two types of stimuli are similar in that they both feature one or more central characters and are designed to elicit positive emotions through their behavior. Previous research has compared animal videos with other video types, finding that dog videos are more effective at improving subjective anxiety, happiness, and positive affect than nature or control videos [17]. In the present study, the humorous human videos elicited levels of positive affect comparable to those elicited by animal videos. However, participants who viewed the animal videos reported stronger character attachment and lower levels of loneliness. These results suggest that animals, compared to humans, more readily evoke attachment in viewers—a mechanism that may more effectively alleviate feelings of loneliness. Why might animals elicit stronger attachment than humans in video contexts? Several explanations are possible. First, positive attitudes toward pets are positively associated with feelings of comfort and pleasure, and interacting with animals can produce a sense of warmth [64]. This aligns with prior findings that physical warmth and social affiliation are interconnected and that experiencing physical warmth increases feelings of interpersonal warmth [65,66]. This sense of warmth may partially underlie the formation of attachment. Supporting this, during the video selection process for Study 2, we found that participants rated animal videos as significantly more “warm” than humorous human videos. These findings suggest that animal videos that convey warmth and emotional healing may be especially conducive to fostering emotional bonds. Second, humans tend to prefer animals they perceive as aesthetically appealing or “cute.” Cuteness is commonly understood as a marker of attractiveness and is associated with the baby schema—a set of infantile features such as large eyes, a round face, and a small nose—that naturally evoke caregiving responses [28]. In this study, the humorous human characters were all adults rather than infants, which likely made the baby schema features in animals more salient by contrast. This, in turn, may have activated stronger caregiving motivations [29]. As a result, the emotional bond formed between humans and animals may closely resemble the adult–infant attachment relationship. Shared features of these two bonds include dependency, proximity-seeking, caregiving behaviors, and affectionate feelings, which together ensure a sense of security, comfort, protection, and survival—for both the child and the animal [28,29,67]. These parallels suggest that the psychological mechanisms underlying cyber-mediated animal attachment are similar to those in infant–caregiver relationships, which may form the emotional foundation through which online animal videos induce attachment and reduce loneliness. Together, these findings suggest that cyber-mediated animal attachment may be facilitated by a combination of affective warmth, anthropomorphic perception, and elicited caregiving responses, processes which may uniquely position animals, more so than humans, as accessible targets of parasocial bonding in digital contexts.

### 4.2. Theoretical and Practical Implications

First, this study introduces the novel concept of cyber-mediated animal attachment and develops a corresponding measurement tool grounded in an established pet attachment scale [55]. This conceptual and methodological advancement offers a new perspective on human–animal bonding, enriching the theoretical frameworks of both pet attachment and virtual attachment.

Second, existing research on short video use has primarily emphasized the adverse consequences of short-video addiction [49,50,51], whereas studies on the potential benefits of animal-related content have largely focused on short-term emotional and stress relief [16,17,18,20,22], with limited attention to the role of psychological bonding. The present study extends this literature by demonstrating that online animal videos can more effectively reduce feelings of loneliness than other types of short videos and that this effect occurs independently of positive emotional responses. These findings offer novel insights into the psychological benefits of digital media and broaden the theoretical scope of research on short-video engagement.

Third, from a practical standpoint, the results of this study not only deepen our theoretical understanding of virtual attachment but also offer valuable guidance for designing digital interventions to mitigate loneliness in an increasingly virtual world. Pet ownership requires various sacrifices, including financial costs and time or energy commitments [68]. It may be especially challenging for young people living in high-pressure, fast-paced environments. In contrast, online animal video engagement does not require actual pet ownership and transcends the limitations of time, space, and cost, making it a low-cost and efficient strategy for alleviating loneliness. The findings suggest that personalized digital interventions based on online pet content could serve as an effective means to promote psychological well-being.

### 4.3. Limitations and Future Research

First, the methodology employed in Study 2 captured only short-term effects of animal video exposure on the development of cyber-mediated animal attachment. The prior literature suggests that the formation of attachment relationships typically requires sustained interaction over time [69]. Consequently, future research could adopt a longitudinal design to examine the long-term impact of online animal video engagement and cyber-mediated animal attachment on loneliness. Additionally, it would be valuable to explore the potential of online animal video as a digital alternative to traditional animal-assisted therapy in delivering enduring psychological benefits.

Second, this study did not explore the boundary conditions that may influence the emergence and effectiveness of cyber-mediated animal attachment. That is, it remains unclear which individuals are more likely to develop strong attachments to cyber-mediated animals and under what circumstances such attachment most effectively promotes psychological well-being. Previous research has indicated that certain personality traits are more strongly associated with internet cat video watching [43]. Future research should consider incorporating moderating variables such as personality traits, emotional needs, or contextual factors to refine our understanding of the antecedents and consequences of cyber-mediated animal attachment. Furthermore, expanding the outcome variables beyond loneliness to include broader well-being indicators would provide a more holistic picture of its psychological implications.

Third, while the current study focused on cyber-mediated animal attachment, it did not investigate its relationship with real pet attachment. Although pet ownership was included as a control variable and was not significantly related to the focal variables, this does not preclude the possibility that meaningful differences—or similarities—exist between attachments to virtual and real-life animals. Future research could directly compare these forms of attachment to elucidate their respective psychological mechanisms, overlap, and distinctions, thereby enriching our theoretical understanding of human–animal interaction in both physical and digital contexts.

Fourth, the participants in the current study is a cultural group who generally holds positive attitudes toward animals [70], and the animals featured in the short videos are often portrayed as cute or anthropomorphized. Consequently, the findings may not generalize to cultures or populations in which animals are associated with negative concepts, such as being dirty, feral, or undesirable [71]. Future research could investigate cross-cultural variations in cyber-mediated animal attachment, exploring how cultural beliefs and attitudes toward animals influence the formation and psychological impact of such attachments.

## 5. Conclusions

This study demonstrates that engagement with online animal videos is associated with reduced feelings of loneliness, and that this relationship is mediated by emotional attachment to the animals portrayed—what we term cyber-mediated animal attachment. By confirming this mediation and ruling out alternative explanations such as personality traits and emotional states, the findings offer robust support for the unique psychological role of virtual human–animal connections. These results contribute to a growing understanding of how parasocial bonds can form not only with human media figures but also with animals in digital contexts. This research highlights the potential of low-cost, scalable digital interventions to address loneliness, particularly for individuals who are unable to own pets due to practical constraints. As digital media continues to shape social and emotional experiences, recognizing and harnessing the affective power of online animal content may offer innovative strategies for enhancing well-being and social connectedness in an increasingly virtual world.

## Figures and Tables

**Figure 1 animals-15-02593-f001:**
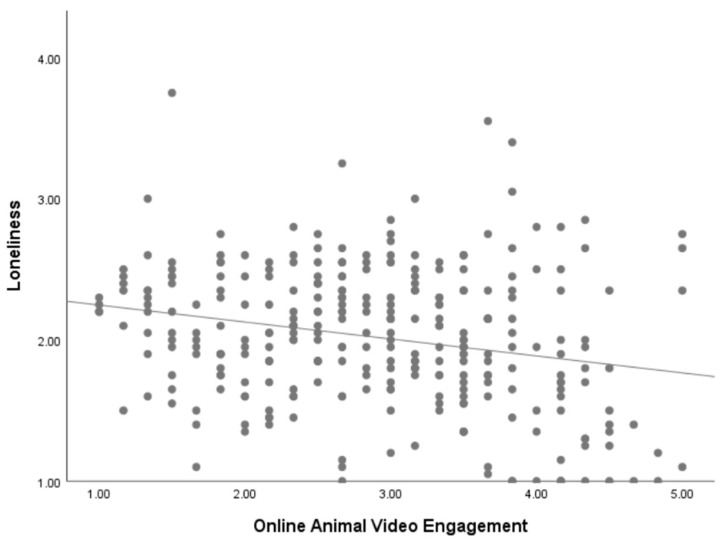
Scatter plot of the relationship between online animal video engagement and loneliness.

**Figure 2 animals-15-02593-f002:**
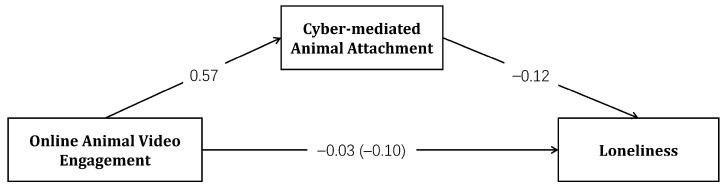
Standardized path coefficients for the mediation model of online animal video engagement, cyber-mediated animal attachment, and loneliness.

**Figure 3 animals-15-02593-f003:**
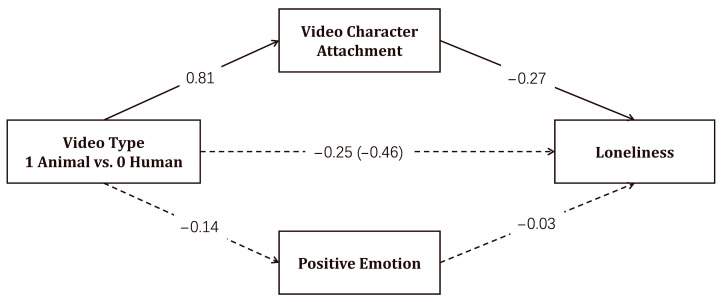
Standardized path coefficients of the mediation model of video type, attachment to video characters, and loneliness.

**Table 1 animals-15-02593-t001:** Descriptive statistics and correlation analysis for key variables (*n* = 299).

	*M ± SD*	1	2	3	4	5	6	7	8
1 Engagement	2.89 ± 0.96	1							
2 Attachment	3.31 ± 0.65	0.61 ***	1						
3 Loneliness	2.02 ± 0.49	−0.23 ***	−0.26 ***	1					
4 Extraversion	3.22 ± 1.04	0.22 ***	0.18 **	−0.48 ***	1				
5 Conscientiousness	3.37 ± 0.86	0.22 ***	0.19 ***	−0.31 ***	0.25 ***	1			
6 Neuroticism	2.63 ± 0.83	−0.12 *	−0.11	0.48 ***	−0.32 ***	−0.37 ***	1		
7 Agreeableness	3.85 ± 0.73	0.11	0.14 *	−0.36 ***	0.18 **	0.30 ***	−0.38 ***	1	
8 Openness	3.48 ± 0.92	0.15 **	0.08	−0.27 ***	0.34 ***	0.29 ***	−0.24 ***	0.25 ***	1

Note: * *p* < 0.05, ** *p* < 0.01, *** *p* < 0.001, the same below.

**Table 2 animals-15-02593-t002:** The mediation analysis for online animal video engagement, cyber-mediated animal attachment, and loneliness (*n* = 299).

Predict Variables	Cyber-Mediated Animal Attachment (Mediator)			Loneliness (Dependent Variable)			Loneliness (Dependent Variable)		
	*b*	*SE*	*t*	*b*	*SE*	*t*	*b*	*SE*	*t*
Engagement	0.39	0.03	11.60 ***	−0.05	0.03	−2.01 *	−0.02	0.03	−0.51
Attachment							−0.09	0.04	−2.05 *
Control Variables									
Sex	0.11	0.06	1.73	0.01	0.05	0.19	0.02	0.05	0.39
Age	−0.02	0.02	−0.78	0.00	0.02	0.18	0.00	0.02	0.09
Pet ownership	−0.04	0.08	−0.59	0.02	0.06	0.34	0.02	0.06	0.27
Personality Traits									
Extraversion	0.03	0.03	0.98	−0.15	0.02	−6.24 ***	−0.15	0.02	−6.15 ***
Conscientiousness	0.04	0.04	1.09	−0.03	0.03	−1.09	−0.03	0.03	−0.96
Neuroticism	−0.01	0.04	−0.19	0.16	0.03	5.13 ***	0.16	0.03	5.13 ***
Agreeableness	0.05	0.05	1.07	−0.11	0.03	−3.14 **	−0.10	0.03	−3.03 **
Openness	−0.04	0.04	−1.05	−0.01	0.03	−0.39	−0.01	0.03	−0.51
*R* ^2^	0.39			0.39			0.40		
*F*	20.26 ***			20.60 ***			19.17 ***		

Note: * *p* < 0.05, ** *p* < 0.01, *** *p* < 0.001, the same below.

**Table 3 animals-15-02593-t003:** The mediation analysis for video type, attachment to video characters, and loneliness (*n* = 131).

Predict Variables	Attachment (Mediator 1)			Positive Emotion (Mediator 2)			Loneliness (Dependent Variable)			Loneliness (Dependent Variable)		
	*b*	*SE*	*t*	*b*	*SE*	*t*	*b*	*SE*	*t*	*b*	*SE*	*t*
Video Type	0.58	0.12	5.01 ***	−0.17	0.21	−0.80	−0.21	0.06	−3.40 ***	−0.11	0.07	−1.72
Attachment										−0.17	0.05	−3.46 ***
Positive Emotion										−0.01	0.03	−0.36
Control Variables												
Sex	0.03	0.12	0.25	−0.11	0.21	−0.56	0.09	0.06	1.54	0.10	0.06	1.68
Age	−0.00	0.01	−0.29	0.00	0.02	0.10	0.01	0.01	0.83	0.00	0.01	0.78
Ownership	−0.16	0.15	1.11	0.33	0.27	1.25	−0.00	0.08	−0.02	−0.03	0.08	0.39
Pre-loneliness	−0.18	0.10	−1.81	−0.31	0.17	−1.80	0.45	0.05	8.68 ***	0.41	0.05	8.30 ***
*R* ^2^	0.21			0.04			0.43			0.49		
*F*	6.63 ***			1.37			18.94 ***			17.11 ***		

Note: Video type can be 1 (animal video group) or 0 (human video group); *** *p* < 0.001.

## Data Availability

The original contributions presented in this study are included in the Appendix A. Further inquiries can be directed to the corresponding authors.

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
