# Peer review of "Animal Video Lovers Always Have Company: The Role of Cyber-Mediated Animal Attachment in Loneliness"

_animals, 2025, doi:10.3390/ani15172593_

Round 1

Reviewer 1 Report

Comments and Suggestions for Authors

The paper analyzes the possibility that a cyber-animal attachment bond can benefit people in terms of reducing feelings of loneliness and creating positive emotional connections.

The work is truly intriguing and original, well-structured, and opens up interesting new perspectives for using short videos with animals to create parasocial bonds. The Big Five is a valid model for personality research, as well as for self-assessment.

Regarding attachment theory with dogs (line 144), it might also be worth mentioning doi 10.1371/journal.pone.0078455

Author Response

The paper analyzes the possibility that a cyber-animal attachment bond can benefit people in terms of reducing feelings of loneliness and creating positive emotional connections.

The work is truly intriguing and original, well-structured, and opens up interesting new perspectives for using short videos with animals to create parasocial bonds. The Big Five is a valid model for personality research, as well as for self-assessment.

Regarding attachment theory with dogs (line 144), it might also be worth mentioning doi 10.1371/journal.pone.0078455

Response: We sincerely thank you for your positive comments on our manuscript, which are truly encouraging.

Regarding the paper you suggested, we have carefully read it and found it very inspiring. We have now cited it in the revised manuscript as reference [35].

Thank you once again for your valuable comments.

Reviewer 2 Report

Comments and Suggestions for Authors

Overall:  Overall, the authors do a good job of explaining the process and hypotheses around this research and their findings.  Additional clarification should be added to the abstract and additional information should also be added to the discussion to bring in cultural context.

Simple Summary and Abstract:  Overall the abstract does a good job of summarizing the study and findings.

Lines 21-22 – Consider rephrasing this sentence to include a phrase that supports the idea that the findings “suggest” that digital animal content could help people rather than stating a definitive. 

Line 38 – The use of the word “clarify” suggests definitive findings in this category.  I recommend rephrasing (“help clarify”).

Introduction:  The introduction does a good job of outlining the study designs and in-depth perspectives of background literature.  It was especially helpful that the authors acknowledged and expanded on the current literature on impact of social media on emotional and social wellbeing and how that may factor into this study.

Study 1:  The authors do a thorough job of describing the methods and results associated with Study 1 with supporting figures and tables. 

Study 2: The authors also did a thorough job describing the methods and results/analysis of study 2 and provided good supporting figures. 

Discussion and Conclusion:  Overall very good.  I appreciate how the authors highlight the importance of compensating for regular social media/video use and how that might factor into the findings. 

The authors do a good job of relating the findings into current research in human animal bonds and attachment theory as underlying factors that could contribute to the findings.  I would recommend that, in this section and potentially within the limitations, the authors also include some additional concepts around culture that may contribute to feelings of attachment.   Some cultures, do not have warm feelings towards animals, specifically dogs and cats, especially if these species are associated with dirty, feral/stray, or other negative concepts.   It would therefore be important to bring into the discussion that the participants were from cultures who have very specific concepts around animals, especially those featured in the videos. 

Author Response

We sincerely thank you for your specific and constructive suggestions, which have been very helpful in improving our manuscript. Our point-by-point responses are provided in the attached document. We would be happy to address any further concerns you may have.

Reviewer 3 Report

Comments and Suggestions for Authors

The present article investigates a topic which is particularly timely, the psychological effects of animal videos. This topic will be of interest to the psychology of internet and to the psychology of human-animal interactions, as well as to the general audience. The article is conceptually clear and very well-written. The hypotheses and analyses are well conceived and well reported. The conclusions are well supported by the results. I have only one major concern regarding the use of the term "cyber-animal" (see below). As for the rest, there are just minor textual corrections.

Line 2: "Have Accompany" --> "Have Company"

Line 13, 27, 34, 41, 69, 78, 106, 131, 133, 142, 175, 178, 185, 206, 209, 214, 218, 221, 225, 227, 238, 239, 247, 272 (two times in the same line), 289, 305, 307, 326, 333, 335, 338, 341, Table 2, Figure 2, 346, 349, 360, 374, 455, 470, 478, 481, 484, 516, 523, 549, 552, 557, 563, 566, 577 and Supplementary Material: "cyber-animal attachment" --> "cyberspace-animal attachment" or "cyber-enabled animal attachment"
The term "cyber-animals" generally refers to animals created through digital animation or robotics. There are also cyber-pets; a famous example is the Tamagotchi, which was popular in the in the 1990s. On the other hand, here you are referring to real biological animals, just seen through cyber-media. Hence, the term cyber-animal should be replaced. An appropriate phrase could be "cyber-mediated animal attachment". Alternatively, since the term cyber-animal is already employed with a different meaning, here you could coin a new term and use "cyberspace-animal" instead of "cyber-animal". Choose the option you prefer between "cyberspace-animal attachment" or cyber-enabled animal attachment", but do not use "cyber-animal attachment". Personally, I think that "cyberspace-animal" would be the best option in this context.

Line 139: "Since secure" --> "Secure"

Line 286: the final full stop is missing

Line 448: "Note: Video Type: 1 animal video group, 0 human video group." --> "Note: Video Type can be 1 (animal video group) or 0 (human video group); *** highlights a p-value < 0.001."

Figure 3: "1 animal" --> 1 Animal" (to be coherent with the rest of the figure, where all the words are capitalized)

Line 524: "in established" --> "in an established"

Line 558: "cyber-animals" --> "cyberspace-animals" (see comment above)

Line 561: "with Internet cat consumption" --> "with internet cat video watching" (in the rest of the paper internet is not capitalized)

Author Response

(The authors gave the same response as above.)

Round 2

Reviewer 3 Report

Comments and Suggestions for Authors

All requests have been appropriately addressed. The phrase "cyber-mediated animal attachment" is perfectly appropriate.